

# Ground-based Multichannel Microwave Radiometer Antenna Pattern Measurement using Solar Observations

Lianfa Lei[1,2,3,4], Zhenhui Wang[1,2], Jiang Qin[3,4], Lei Zhu[3,4], Rui Chen[3,4], Jianping Lu[3,4], Yingying Ma[5]

[1] Collaborative Innovation Center on Forecast and Evaluation of Meteorological Disasters, CMA Key Laboratory of Aerosol-Cloud-Precipitation, Nanjing University of Information Science & Technology, Nanjing 210044, PR China;

[2] School of Atmospheric Physics, Nanjing University of Information Science & Technology, Nanjing 210044, PR China;

[3] North Sky-Dome Information Technology(Xi'an) CO., LTD, Xi'an 710100, PR China

[4] Xi'an Electronic Engineering Research Institute, Xi'an 710100, PR China

[5] State Key Laboratory of Information Engineering in Surveying, Mapping and Remote Sensing (LIESMARS), Wuhan University, Wuhan 430074, PR China

*Correspondence to*: Zhenhui Wang (eiap@nuist.edu.cn)

**Abstract.** Ground-based multichannel microwave radiometers (MWRs) can provide continuous temperature and humidity profiles of the troposphere. MWR antenna pattern measurements are important for reliable and accurate antenna temperature measurement and are usually carried out in a microwave anechoic chamber. Measurement using an anechoic chamber is complex and expensive because the conventional measurement procedure requires a special situation and professional instruments. More importantly, the construction of the anechoic chamber and the installation method of the absorbing material can directly influence the performance of the anechoic chamber and the result of the antenna measurement. This paper proposes a new method of MWR antenna measurement by observing the sun, and this method can be used to measure other radar antenna patterns. During the measurement, the MWR observes the microwave radiation brightness temperature (TB) to measure the antenna pattern by high-resolution raster scanning of the azimuth and elevation of the sun under a clear sky in Xi'an, China. Analysis of the TB scanning data of the sun at four frequencies, 22.235, 26.235, 30.000 and 51.250 GHz, showed that the microwave radiation TB of the sun is strong enough to be observed by the MWR. Furthermore, the antenna pattern was illustrated and analyzed based on these data, which fully proves that the sun can be used to measure the antenna pattern. Finally, the antenna pattern derived from the solar observation was compared with the result of the far-field measurement with a point source in the microwave anechoic chamber at 30 GHz, the maximum error of the beamwidth is less than 0.1°, which showed that this pattern matched well to the pattern measurement using a point source in the microwave anechoic chamber. Therefore, the antenna pattern of the MWR can be measured by scanning the sun without a point source in the microwave anechoic chamber, and this method can be used for convenient MWR antenna measurements and can reduce the measurement complexity and cost.

## 1 Introduction

The MWR can continuously provide temperature, water vapor, cloud liquid and humidity profiles up to



10 km height through a neural network algorithm, it is based on the theory of atmospheric microwave
thermal radiation and its transfer in the atmosphere, atmospheric radiation TB at some frequencies in K-
and V-bands that derive from atmospheric water vapor, cloud liquid water and molecular oxygen
emissions can be observed. At present, MWRs are mainly used to remotely sensing the atmospheric
temperature and humidity profiles in the troposphere, many researchers have proved that an MWR can
provide valuable data on the temperature and humidity structures of the troposphere (Cadeddu et al.,
2013; Ahn et al., 2016; Laura et al., 2017). Furthermore, there are many other applications of MWRs, for
example, nowcasting of heavy rain events using microwave radiometer has been carried out at Kolkata
(Chakraborty et al., 2014). He et al. (2020) studied the influence of assimilating MWR data into the WRF
precipitation model. In addition, Wang et al. (2014) presented the theoretical research for the lightning
TB response of an MWR, and Jiang et al. (2018) remotely sensed artificially triggered lightning events
with an MWR and researched the lighting heating radiation of microwave and the effect duration-this is a
new application.
MWRs are not only used to remotely sensed temperature and humidity profiles in the tropospheric
atmosphere but are also used to observe the microwave radiation of the sun. Solar radiation is very strong,
and the sun is a good and far enough point target to be used for antenna pattern measurement. Existing
research shows that strong cosmic radio sources (such as the sun, the moon, Mars, etc.) can be used to not
only calibrate radar antenna pointing but also measure antenna patterns. Several researchers used
previously established results of the sun and cosmic sources to measure radar antenna patterns
(Huuskonen, et al., 2007; Jin, et al., 2010; Altube, et al., 2016; Holleman, et al., 2010; Reimann, et al.,
2016). In addition, Ulich (1977) measured the gain of a microwave antenna with a radiometric method,
and an absolute uncertainty less than 0.1 dB was achievable.
The applications of MWR are increasing widely. The traditional far-field method of measuring the
antenna pattern is complex with a point source in an anechoic chamber, as the traditional method needs
an artificial calibration reference, special situation and no reflections from the ground and surrounding.
Furthermore, when the antenna is very large or when the final stages of assembly occur at the installation
site, the traditional method is extremely difficult (Johnson, et al., 1973). For those and other reasons, we
need to adopt a new and simple method to measure the antenna pattern. A good estimation of the real
antenna pattern can be derived using the sun, and it is rare to observe solar radiation with an MWR to
study the solar characteristics and to measure the MWR antenna pattern. This new method does not need



to consider the influence of ground reflection and the environment and does not need to be conducted in
an anechoic chamber. This paper proposes observing the sun to measure the MWR antenna pattern, we
will improve and perfect the measurement method of observing the sun for both theoretical and practical
application.

This paper presents an experiment in which the sun was remotely sensed with an MWR at a Xi'an field

experimental site (N34.091°, E108.89°), dedicated raster scans of the sun were used to derive the antenna
pattern. This paper is divided into four sections: Section 1 introduces the background, significance of this
research and the solar observation scheme based on the MWR. Section 2 discusses the solar TB response
model based on the transmission theory of thermal radiation in the microwave band and the MWR
antenna measurement method. Section 3 explains the MWR data obtained in solar observation mode and
then analyzes the amplitude of the TB increment and the measurement results. The scan strategy and
signal processing are described in detail. Section 4 is a summary and discussion of the experimental
results. Finally, the experimental results show that the method of the MWR antenna measurement based
on solar observations has the advantages of simple operation and high accuracy.
**2 Instrument**
The MWR (model MWP967KV) used for this experiment, shown in Fig. 1, was developed and
manufactured by our research team, and the system performance parameters are given in Table 1. The
MWR observes radiation intensity in K- and V-bands in order to obtain the atmospheric temperature,
humidity, cloud water, etc. It comprises an antenna, two sensitive receivers, a detector unit and a data
retrieval system, its antenna beamwidth is about 1.9±0.8° at 51–59 GHz and approximately 3.8±0.8° at
22–30 GHz. The MWR also contains a high-precision elevation and azimuth stepping scanning system to
scan the sky, and the angular resolution is 0.1°for both the elevation and azimuth. At present, the MWR is
widely used in atmospheric observations.

To ensure high observation accuracy of atmospheric radiation, calibration of the MWR is necessary. It

is calibrated by liquid nitrogen (LN2), hot load, noise diode and the Tipping curve method (Han and
Westwater, 2000). These calibrations can provide an absolute accuracy of at least 0.5 K (Li, et al., 2014).
When the MWR is in the meteorological observation mode, the TB data of 22 frequencies are observed
by default in the frequency range of 22-59 GHz. Frequencies between 22 and 30 GHz are mostly





sensitive to atmospheric water vapor and liquid, and frequencies between 51 and 59 GHz are sensitive to
atmospheric temperature due to the absorption of atmospheric oxygen.

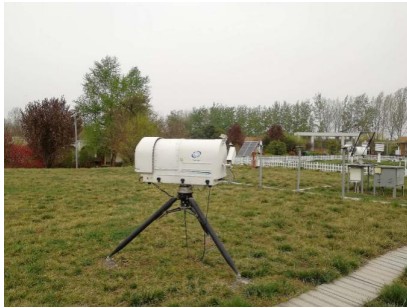


**Figure 1.** The ground-based multichannel microwave radiometer used in this experiment.

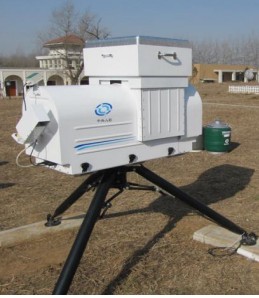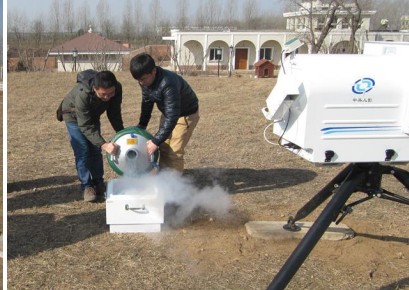

**Figure 2.** The liquid nitrogen (LN2) calibration.
Table 1 The system performance of the MWR used in this study

| Parameter | Specification |
|---|---|
| Output brightness temperature | K-band: 8; V-band: 14 |
| Channel frequency | K-band: 22-30 GHz; V-band: 51-59 GHz |
| Brightness temperature range | 0-800 K |
| Brightness temperature accuracy | 0.5 K |
| Brightness temperature resolution | K-band: ≤ 0.2 K (RMS); V-band: ≤ 0.3 K (RMS) |
| Integration time | Typically 1s |
| Antenna scanning capability | +/-180° stepping scanning; |
| Antenna angular resolution | 0.1° |
| Beam width | K-band: 3.8±0.8°; V-band: typically 1.9±0.8° |
| Calibration method | Hot load; Noise diode; Tipping; LN2 |
| Working voltage | AC 220 V±20 %, 50Hz±5 % |
| Dimensions | 1262×615×457 mm |
| Power | Normal steady state < 1.5 A |
| Working environment | Temp: -40-50 °C; RH: 0-100 %; |
| Weight | Approximately 80 kg |



## 3 The sun and the mode of MWR observations

The quiet sun has relatively quiet and stable radiation in the microwave radio spectrum, and the general activity at the low end of the radio frequency spectrum is much greater than that in the microwave region(Aarons, 1954). The microwave noise of the sun is preferred as a source to measure the antenna pattern, and the microwave radiation of the sun can be detected through clouds.

To use the MWR to scan the sun, we have to know the position of the sun. Although the solar system is a complex moving system, we can accurately calculate the position of the sun because of the relative motion between the sun and the earth. The formulas for the elevation and azimuth calculation of the sun are as follows (Qi, et al., 2019):

$$E_l = sin^{-1} sin(\theta_{lat})sin(\delta) + cos(\theta_{lat}) cos(\delta)cos(T_0) \tag{1}$$

$$A_z = cos^{-1}\left[\frac{sin(E_l)sin(\theta_{lat})-sin(\delta)}{cos(E_l)cos(\theta_{lat})}\right] \tag{2}$$

$$T_0 = (t_s - 12) \cdot 15^{\circ}, \tag{3}$$

where $E_l$ and $A_z$ are the elevation and azimuth of the sun, respectively. $\delta$ is the declination of the sun, $\theta_{lat}$ is the latitude of the MWR and $t_s$ is the hour angle of the sun. The sun can be assumed to be in the far-field region for the antenna and it can be treated as a homogeneous disc with an approximate diameter of 0.53°. It is widely accepted that the sun can be used for measurement of the antenna pattern.

When the MWR is in the solar observation mode, the antenna system will be adjusted to control its antenna beam pointing to the position of the sun and tuning the antenna beam to scan the sun's azimuth and elevation within a range of ±10° of the sun position and to observe the TB. The maximum amplitude of the TB is received by the MWR when the antenna beam points to the center of the sun. Then, the azimuth and elevation biases of the antenna beam between the peak TB and the predicted sun position are the corresponding calibration angle. This method does not need any additional artificial calibration reference because of the solar position can be accurately predicted. The result of this process is the measurement of the antenna temperature due to the sun. The MWR is made using a small antenna, and its beam solid angle is larger than the solar disk. Fig. 3 shows the spatial relationship between the solar and the antenna beam.





Figure 3. Antenna beam scanning the solar disk.

To further improve the pointing accuracy of the antenna and to decrease the influence of atmospheric

refraction during the propagation of electromagnetic waves, atmospheric refraction correction cannot be

ignored when the elevation and azimuth angles are calculated. Huuskonen and Holleman (2007)

presented an atmospheric refractivity correction method that is accurate enough for the measurement of

antenna calibration at low elevation angles. As the atmospheric refraction and attenuation both depend on

elevation pointing, these effects have to be taken into account because solar radiation travels farther in

the atmosphere at lower elevations.

**4 Theoretical analysis**

**4.1 The model of atmospheric TB**

Due to the limited sensitivity and a large beamwidth, only solar emissions can be easily detected with the

MWR. The response of the MWR to solar radiation is studied in order to measure the antenna pattern.

The brightness temperature can be estimated by the radiative transfer equation through the atmosphere.

When the antenna scans the sun, the TB received by the antenna can be determined as follows (Coates,

1958; D'Orazio, et al., 2003):

$$T'_{sun}(\varphi,\theta) = \frac{\varepsilon}{\Omega_A} \iint_{\Omega} G(\zeta,\xi)\{T_{bg}\,e^{-\tau(\theta)} + T_m[1 - e^{-\tau(\theta)}] + T_{sun}e^{-\tau(\theta)}\}d\Omega \qquad (4)$$

$$\Omega_A = \iint_{4\pi} G(\zeta,\xi)d\Omega. \qquad (5)$$

When the antenna scans the sky, the sun is not in the beam, and the atmospheric TB can be calculated

as follows:

$$T'_{sky}(\varphi,\theta) = \frac{\varepsilon}{\Omega_A} \iint_{4\pi} G(\zeta,\xi)\{T_{bg}\,e^{-\tau(\theta)} + T_m[1 - e^{-\tau(\theta)}]\}d\Omega, \qquad (6)$$



where $\varepsilon$ and $G$ are the respective transmission and gain of the antenna, $\varphi, \theta$ are the respective azimuth
and elevation angle, $\zeta$ and $\xi$ are the respective azimuth and elevation angle radial distance from the
beam center, $\Omega$ is the sold angle, $\Omega_A$ is the antenna sold angle, $T_m$ and $T_{bg}$ are the respective
atmospheric effective temperature and cosmic background brightness temperature ($T_{bg}$ = 2.75 K), $T_{sun}$ is
the average TB of the sun and $\tau(\theta)$ is the atmospheric opacity of each direction.

Under a clear sky, ignoring the radiation bending caused by the refractive index gradient in the

plane-stratified atmosphere, the opacity is given as follows (Ulich, et al.,1980; Xie, et al., 2013):
$$\tau(\theta) = \tau(90°)M(\theta) \qquad (7)$$
$$M(\theta) = 1/\sin(\theta), \qquad (8)$$
where M is the air mass calculated as a function of the antenna elevation, and the link between the sky TB
and the atmospheric opacity is given as follows (D'Orazio, et al., 2003; Han and Westwater, 2000; Zhang,
et al., 2016):
$$\tau(\theta) = \ln\left[\frac{T_m - T_{bg}}{T_m - T_b'(\theta)}\right]. \qquad (9)$$

Thus, the atmospheric opacity $\tau(90°)$ at the zenith is obtained from the slope of a plot of $\tau(\theta)$ vs.

$M(\theta)$. A typical plot is shown in Fig. 4.

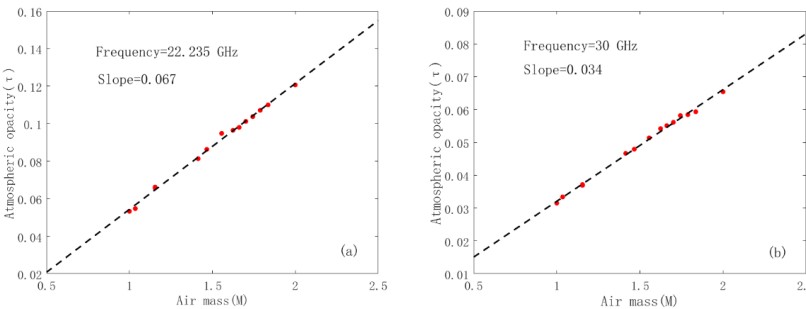

**Figure 4.** Plot of atmospheric opacity vs air mass at 13:10(local time) on Feb. 21, 2020.
(a) 22.235 and (b) 30.000 GHz.
**4.2 The model of the antenna power pattern**
In this paper, the azimuth and elevation scanning are used to derive and measure the antenna pattern.
When the antenna is based on azimuth scanning, the respective solar azimuth and elevation angle are $\varphi_s$
and $\theta_s$, and the elevation angle of the MWR is equal to the solar elevation angle $\theta_s$ when the solar is in
the center of the antenna beam. When the antenna is used for elevation scanning of the sky, the azimuth
of the antenna is a constant $\varphi_0(|\varphi_0 - \varphi_s| \geq 10°)$.



In the series of sun scans just described, the antenna position is fixed during each sun scan, and a
recording is made of both the sun and the sky at the same elevation. Atmospheric opacity can be
considered constant, and the TB increment of the solar radiation arriving at the antenna without
atmospheric attenuation can be obtained by subtracting (6) from (4):
$$\Delta T_{sun}(x,y) = \frac{\varepsilon}{\Omega_A} \iint_{\Omega} G(\zeta, \xi)\, T_{sun} d\Omega \tag{10}$$
$$P(x,y) = \frac{\varepsilon}{\Omega_A} T_{sun} \Gamma(x,y), \tag{11}$$
where P is the TB increment because of the solar radiance, and it is the received solar power as a
function of the angle radius from the center of the beam, $x$ and $y$ represents the angle radius distance
from the sun to the center of the antenna beam. Because of the directional characteristics of the antenna,
if the antenna is completely surrounded by a source, then the antenna temperature will be equal to the
brightness temperature of the source, when the source does not completely surround the antenna, then the
ratio between the antenna temperature and the brightness temperature is equal to the ratio of the integral
of the antenna pattern over the source to the integral of the antenna pattern over a $4\pi$ solid angle (Coates,

1958).

Assuming a uniform disk for the sun, the simplest reasonable model of the actual antenna power
pattern is the Gaussian function (Ulich and Haast, 1976; Holleman, 2010), the power pattern $P(x,y)$ is
given by
$$P(x,y) = A_d \exp\left[-4\ln 2\left(\frac{x^2}{\theta_H^2} + \frac{y^2}{\theta_E^2}\right)\right] \tag{12}$$
$$G(x,y) = \exp\left[-4\ln 2\left(\frac{x^2}{\theta_H^2} + \frac{y^2}{\theta_E^2}\right)\right], \tag{13}$$
where $A_d$ is the TB increment of the antenna beam pointing to the center of the sun, and $\theta_H$ and $\theta_E$ are
the half-power beamwidth of the H-plane and E-plane, respectively. $G(x,y)$ is the normalized antenna
2-dimensional power pattern.
The sun is known to be a good target for MWR and we can use the sun to measure the antenna pattern.
In this method, The MWR adopts the stable elevation-over-azimuth antenna positioner system and the
MWR antenna beam is controlled to scan the sun. When the MWR antenna scans the sun, a rotation of
the antenna in azimuth with constant elevation is not a large circle in the sky-sphere and there are some
distortions in the scanning path. The distortion is small at low elevation scanning, but it cannot be
ignored at high elevation. There is an extreme case when the azimuth of the antenna is rotated at an



elevation angle of 90°, but the antenna beam will not move in the sky (Reimann and Hagen, 2016). In
order to calibrate this distortion, we need to adopt an accurate method, Reimann and Hagen (2016)
described the mathematical statement and calculation method in detail. This method is used to calibrate
the angle distortion in our observation throughout the presented analysis.
We used the MWR to track and scan the solar at some frequencies on sunny days. Measured TB
increment distributions for a 29 by 29 matrix of measurement scatterplots and corrected the angle
distortion are shown in Fig.5. There are more raster scanning points near the sun and fewer points away
from the sun, and this scanning method will help to get the complete 3-D antenna pattern. The solar
power data exhibit a radial pattern with a clear maximum in the center, this maximum value and the size
of the halo are different for each frequency, and they are related to the antenna beamwidth.

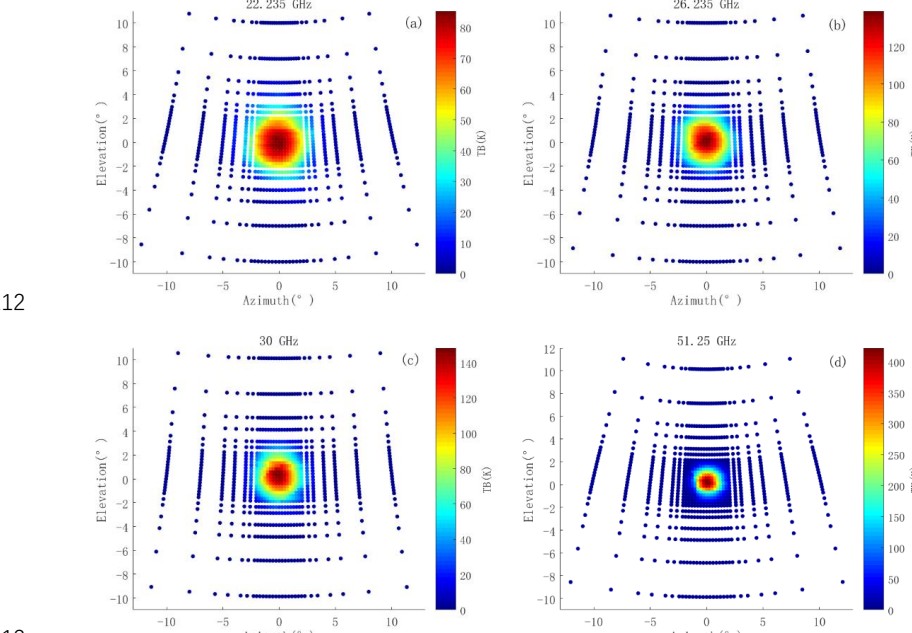



**Figure 5.** Two-dimensional distribution of the TB increment observed from the sun raster scanning on March 14,
2020, at four typical frequencies. The vertical axis is the difference between the antenna elevation and the elevation
of the sun, and the horizontal axis is the same for the azimuth. (a) 22.235, (b) 26.235, (c)30.000, and (d) 51.250 GHz.
Figure 6 shows the results from fitting the Gaussian model, and the 3-D pattern of the antenna can be
obtained by least squares fitting of the antenna raster scanning data. In this way, the beamwidth of the
antenna can be obtained.



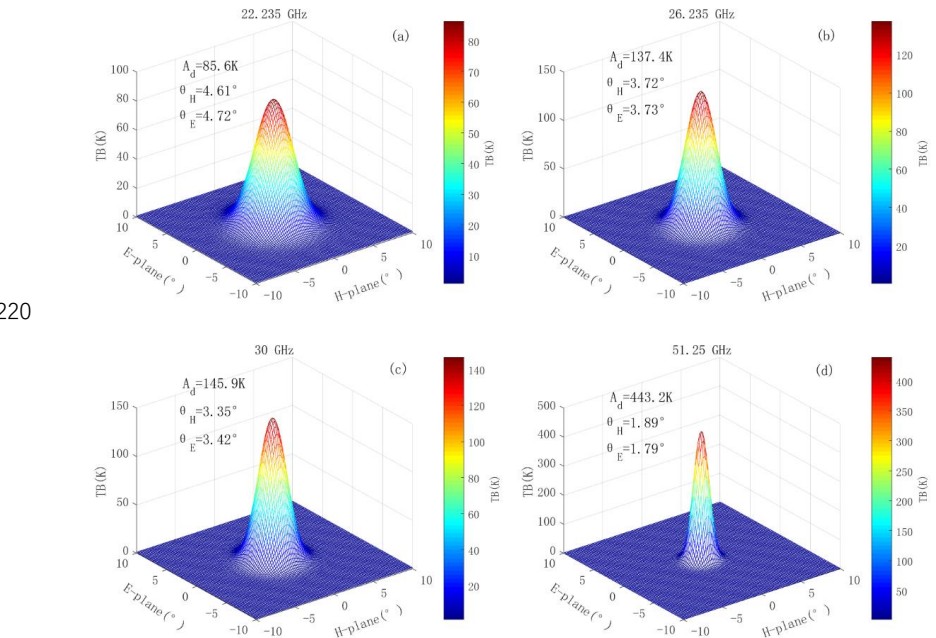



**Figure 6.** The 3-D antenna power pattern at four frequencies. (a) 22.235 GHz, (b)26.235 GHz, (c)30.000 GHz and (d) 51.250 GHz.

The antenna pattern was measured and analyzed with the above Gaussian antenna pattern. From the observation results, the antenna power pattern is relatively symmetrical, the channel with a lower frequency has a wide beamwidth and a small amplitude of the TB increment, while the channel with a higher frequency has a narrow beamwidth and a large amplitude of the TB increment (Fig. 6). The measured value of beamwidth shows good agreement with the theoretical simulation value and complies with the design specification. Hence, a good estimation of the real antenna pattern can be derived using the sun. The azimuth and elevation cut position for the sun is chosen to be the expected main beam direction (Fig.7).

Because of the strong absorption of the atmosphere, when the microwave radiation of the solar is absorbed by the atmosphere and cannot reach the earth's surface at some frequencies in the V-band, the antenna power pattern cannot be measured at these frequencies.





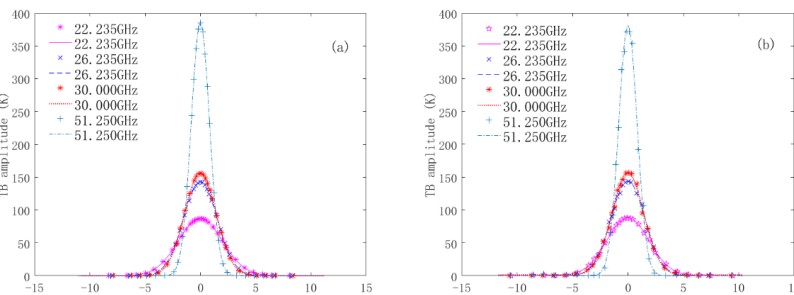


**Figure 7.** The antenna power pattern. (a)H-plane, and (b)E-plane.
Generally, the antenna power pattern is measured in at least two principal planes (H- and E-plane).
Therefore, we only observed and analyzed the solar azimuth and elevation scanning data to determine the
beamwidth of the antenna. Besides the sun measurement, a point source measurement was performed for
comparison, and a signal generator was used to provide a known and constant signal source to measure
the antenna pattern in the microwave anechoic chamber. We measured the antenna pattern at 30 GHz, the
pattern derived from the solar and a point source were compared and the maximum error was less than
0.1° at 30 GHz (Fig. 8). It is shown that the main-lobe matches well to the pattern based the point source
measurements. The result shows that when the antenna gain decreases more than 25 dB, the MWR
cannot measure the TB increment from the sun.

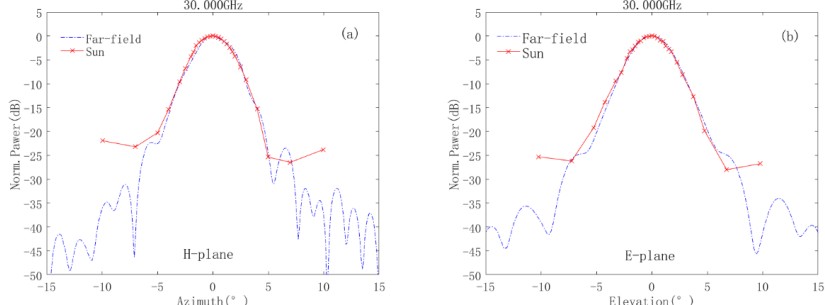


**Figure 8.** The antenna patterns measured in the anechoic chamber as compared with the results from the scanning of
the sun at 30GHz. (a)H- plane and (b)E-plane.
**4.3 The calibration method of antenna gain**
Several methods are commonly used for experimentally determining the peak power gain of an antenna,
each technique has its own particular advantages and disadvantages. The antenna maximum direction
gain can be easily calculated by using the following formula when we have the beamwidth of the antenna





(Ulich,1977):

$$D = \frac{4\pi R}{\Omega_D C_r},$$      (14)

where $\Omega_D$ is the solid angle of the solar, and $C_r$ is the correction factor that accounts for the partial
resolution of the disk image by the main lobe of the antenna pattern. We can ignore other effects, and $C_r$
can be closely approximated by

$$C_r = \frac{1 - \exp\left[-ln2(\theta_D/\theta_A)^2\right]}{ln2(\theta_D/\theta_A)^2}, \quad if\ \theta_D < \theta_A.$$      (15)

The ratio R is defined as follows:

$$R = \frac{\iint_{\Omega_D} G d\Omega}{\iint_{4\pi} G d\Omega}.$$      (16)

At the same time, the antenna effective area $A_e$ and the aperture efficiency $\eta$ are calculated from
(Holleman, 2010)

$$A_e = \frac{\lambda^2 D}{4\pi}$$      (17)

$$\eta = A_e/A_g * 100,$$      (18)

The effective area of the antenna is typically around 50%–60% of the physical area, and it is usually
very difficult to determine (Tapping, 2013). However, it can be calculated by observing the sun. The
calcsulated parameters of the antenna pattern are shown in Table 2.
During the observation, the TB increment of the solar received by the MWR was recorded at four
frequencies. The antenna system of MWR has a narrow beamwidth and larger TB increment at high
frequency. The calculated gain is greater than 30 dB each frequency, and the MWR aperture efficiency is
around 45%–50%. Compared with the beamwidth of H-plane and E-plane, it can be seen that the
beamwidth is similar, and the antenna beam is approximately a circular beam.

Table 2 The summary of the MWR antenna parameters

| Frequency(GHz) | $\Delta$Tsun(K) | Gain(dB) | $\eta$(%) | Beamwidth(°) H-plane | Beamwidth(°) E-plane |
|---|---|---|---|---|---|
| 22.235 | 85.6 | 32.6 | 45.1 | 4.61 | 4.72 |
| 26.235 | 137.4 | 34.5 | 50.4 | 3.72 | 3.73 |
| 30.000 | 145.9 | 35.3 | 46.8 | 3.35 | 3.42 |
| 51.250 | 443.2 | 40.7 | 54.3 | 1.89 | 1.79 |

**5 Conclusions**
This paper presents a novel method to measure the antenna pattern by observing the sun, this method



overcomes the operational complexities and the disadvantages of high costs in the traditional method. In
order to measure the MWR antenna pattern with the sun, the tracking and scanning sun observation
experiment was carried out in the Xi'an field experiment by using the MWR.
It has been shown with the observed data that the MWR can respond to the solar microwave radiation,
and the TB increment can be observed when the sun is in the antenna beam. The sun is used as a point
source, and the 3-D antenna pattern can be measured by raster scanning of the sun. During the
measurement, the MWR antenna gain exceeds 30 dB and 40 dB in the K-band and V-band, the antenna
aperture efficiency is about 45%–55%, the beamwidth is about 3–5° in the K-band and it is less than 2° in
the V-band. A comparison between the sun and a point source measurement showed an agreement with
the beamwidth in both azimuth and elevation scanning, the maximum error was less than 0.1° at 30GHz,
the antenna main lobe pattern can be measured completely and the TB increment of the sun could not be
received on the sidelobe because the sidelobe gain is too small. This paper shows that a good estimation
of the real antenna pattern can be derived using the sun, and this study presents the MWR antenna beam
measurement method using the sun as a radiation source to quickly measure and calculate beamwidth,
gain and the effective antenna area. Therefore, the sun as a radiation source can be widely used for MWR
antenna pattern measurement, and this will greatly simplify the operation process of antenna pattern
measurements.
The method provided in this paper can be used as a reference for antenna pattern measurements by
making use of the sun. An important advantage of this method is that it is simple to employ, with easily
achievable goals, and it is very suitable for the antenna pattern measurement in some cases when the
antenna cannot be moved from its operating environment, or is staged at the installation site. In the future,
we can not only use the sun to measure the antenna pattern, but also use the MWR to observe the
microwave radiation of the sun.
**Acknowledgements.** This work was supported by the National Natural Science Foundation of China
(NO.41675028), and A Project Funded by the Priority Academic Program Development of Jiangsu
Higher Education Institutions [PAPD] and the Natural Science Foundation of Shanxi Province, China
(NO.2020JM-718). We are also grateful to Shanxi Provincial Atmospheric Sounding Technical Support
Center for the radiometer installation and for their support.



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
