# Peer review of "Ground-based Multichannel Microwave Radiometer"

_Atmospheric Measurement Techniques, 2020_

## Referee Comment (RC1) · Anonymous Referee #1 · 28 Sep 2020

**Review of the manuscript "Ground-based Multichannel Microwave radiometer Antenna Pattern Measurements using Solar Observations" by Lianfa Lei et al.**

The manuscript presents an estimation of the amplitude antenna pattern using solar scans. The authors show an application of the method to solar measurements from a multi-channel microwave radiometer. A good agreement of the retrieved patterns with those measured in an anechoic chamber was found.

In my opinion the manuscript has a number of flaws:

1. Lack of novelty. The presented method has been known for long time. Antenna pattern measurements with the sun as a signal source are widely used for active and passive microwave instruments in meteorology (Reimann and Hagen, 2016 "Antenna Pattern Measurements of Weather Radars Using the Sun and a Point Source" and references therein). In particular, solar scans are recommended as a daily routine for weather radars of national weather services (Chandrasekar et al 2015 "Calibration Procedures for Global Precipitation-Measurement Ground-Validation Radars", Fresch et al 2019, "Pointing Accuracy of an Operational Polarimetric Weather Radar"). Authors themselves give a list of references confirming this.

2. A large part of the manuscript shows radiometry and antenna basics, which can be found in every handbook on the microwave radiometers (e.g. in Ulaby and Long "Microwave Radar and Radiometric Remote Sensing").

3. The authors do not mention design and dimensions of the antennas. Typically, mirrors with no subreflectors are used. In this case the antenna properties can be relatively accurately approximated following known relations between the gain and beamwidth and the antenna aperture size. It is also not clear why uncertainties of the beamwidth are so large (given are 3.8+/-0.8 deg and 1.9+/-0.8 deg). If the design of the antenna system is known, it should be possible to calculate the beam width with an accuracy of 15 %.

4. Lack of motivation for meteorological applications of MWR. For main applications of radiometers in meteorology, which are temperature and humidity profiling and integrated water vapor, the atmosphere is often assumed to be uniform, i.e. the radiation is constant within the antenna beam. Taking into account that beamwidths of MWR are in the order of 1-4 deg, in the case of clear sky this assumption is fully justified. In such cases only the antenna loss, which is included in the total loss during the hot-cold calibration, matters. Gain and beamwidth have no impact. In presence of liquid clouds, the radiation within the beam is not homogeneous. But it is not clear how more precise knowledge of the beamwidth and gain can help to take this effect into account.

Taking the above-mentioned points, I think that the manuscript does not fulfil requirements for the publication in AMT and should be rejected.

---

## Referee Comment (RC2) · Anonymous Referee #2 · 6 Oct 2020

**Review for **"Ground-based Multichannel Microwave Radiometer Antenna Pattern Measurement using Solar Observations"** by Lei et al., submitted to Atmospheric Measurement Techniques**

**Synthesis:**

This paper presents a method to determine microwave antenna patterns from solar scans instead of using an anechoic chamber. The authors apply a scan pattern a ground-based atmospheric microwave radiometer. The results prove to be reasonably accurate by comparing to laboratory measurements.

**General comments:**

I am missing some motivation for the study: Of course, it is always desired to characterize an instrument as well as possible. However, more motivation on why is it important to know the exact antenna patterns for ground-based microwave radiometers would be necessary to mention (e.g. temperature profiling, considering antenna patterns in radiative transfer modelling for retrieval algorithms, etc.)

The sun has been used as target for antenna pattern studies before. Please make clear what your novelty is. Would it be possible to apply your method automatically to check the pointing accuracy as well as the alignment of a radiometer that is deployed in the field?

The basics on microwave radiometry and antennas cover too much of the manuscript. Most of the information can be found in textbooks. On the other hand, the technical description of the instrument is very poor in terms of receiver technique and other components. How is elevation and azimuth scanning performed by the instrument? What are the sources of uncertainty in this regard? What is the temporal resolution of the brightness temperature measurements when performing scans?

The experimental design was not described in a reproducible way, there is a lot of practical information missing. How much time does one scan pattern take? And more in detail: When did you perform the measurements that this manuscript is based on? Did you combine several scan patterns for your results? Under which solar elevation angles were the scans performed? How did you consider the movement of the sun's position during one scan?

What about the repeatability of the method? Did you perform several scans under different conditions and/or solar elevation angles (e.g. summer/winter, morning/noon)? What are your recommendations in that respect?

**Summary**

To summarize, I cannot recommend publication of this manuscript at this stage. The manuscript needs major revisions by carefully considering all the above-mentioned points.

---

## Referee Comment (RC3) · Anonymous Referee #3 · 7 Oct 2020

General remarks

The manuscript describes antenna measurements of a microwave radiometer using the sun. To my knowledge this is one of the first thoroughly study done for temperature and humidity microwave profiler. However, antenna measurements using the sun is a well-established technique and e.g. applied routinely by the European weather radar community for monitoring receiver performance and pointing accuracy. I miss any clear statement why it is necessary for microwave radiometer measurements and what are the benefits of such measurements. In some points the manuscript gives unnecessary details.

[Figure]

In the present state the manuscript requires major revisions before it can be published.

- The method is not novel, nor is the application novel. A potential novelty needs to be elaborated in detail and the benefit for MWR measurements have to be highlighted

- Technical detail of the MWR model are missing: antenna size, frequencies in K- and V-band, number of receivers

- Equations for sun azimuth and elevation are not necessary since they don't describe the necessary information as long as the declination is not given

- Too many details in the sections about "model of atmospheric TB" and "model of the antenna power pattern". Why is the opacity of the atmosphere relevant for this study?

Minor remarks

English grammar and spelling need to be checked. There are many recurrences of statements.

The DOI links in the references have wrong syntax: should be https://doi.org/ or doi: instead of https://doi:

Lines 38 - 39: check English grammar

Line 44: Laura et al., 2017 is missing in references

Lines 48 - 50: check English grammar

Lines 51 - 71: several recurrences of statements

Figure 2 is not necessary, this is not a paper about calibration

Figure 3 is not necessary, the solid angle of the sun and estimated solid angle of the antenna give sufficient information to the reader

Line 140: "only solar emission" What else?

Line 152: "solid" instead of "sold"
Figure 4: different date as in Figure 5. How can this be related to each other? Not relevant for this manuscript

Lines 170 - 175: very difficult to understand. How can the antenna be fixed (line 174) during a sun scan (line 174)?

Line 173: smaller than 10°

Lines 182 - 187: I do not understand the context

Line 189: Holleman et al., 2010

Line 196: Remove, this has been written very often now

Lines 217 - 219: what is the calibration angle? Is it 0 in azimuth and 0 in elevation?

Figure 7: what are the markers, what are the lines?

Line 237: what is the H-plane and E-plane in this context? Solar radiation is not polarized

Line 254: what is D?

Line 262: Holleman et al., 2010

Line 267: calculated

Table 2: it would be necessary to have more than one measurement series to get statistical relevant accuracy of the antenna parameters

Line 297: Why should one use a MWR to observe the radiation of the sun? What will be the context? What will be the difference to the current manuscript?

---

## Author Comment (AC1) · 24 Nov 2020

Thank you very much for your letter and suggestion.

The comments are all valuable and very helpful for revising and improving our paper. We have studied comments carefully, and have made corrections and modification which we hope meet with approval. We have been trying our best to improve the manuscript. Please see the revised version of the manuscript for detail. The following is the correspondence to the comments.

1. Comment: Lack of novelty. The presented method has been known for long time.

[Figure]

Antenna pattern measurements with the sun as a signal source are widely used for active and passive microwave instruments in meteorology.

Response: Thanks for your suggestion, may be the former version didn't describe the innovative points prominently. We have noticed that several researchers in literature used the sun to measure weather radar antenna pattern. In this paper, our study is to accurately measure the antenna pattern of MWR with a simplified solar method. We apply this method to measure the MWR pattern and this method can be used to improve the Tipping curve calibration accuracy, automatically check and calibrate the pointing accuracy. Therefore, this method can be used for MWR antenna measurements and hopefully to monitor the antenna pattern and pointing of a radiometer in operational, field applications after installation. The description above has been added to the manuscript. Please see Lines 39 - 57 on Pages 1 - 2 of the revised version.

2. Comment: A large part of the manuscript shows radiometry and antenna basics, which can be found in every handbook on the microwave radiometers.

Response: Thank you for your suggestions. We have simplified the part about radiometry and antenna. Furthermore, we have added the antenna design description and the schematic antenna structure to the manuscript. Please see Pages 2 - 4 of the revised version.

3. Comment: The authors do not mention design and dimensions of the antennas. Typically, mirrors with no subreflectors are used. In this case the antenna properties can be relatively accurately approximated following known relations between the gain and beamwidth and the antenna aperture size. It is also not clear why uncertainties of the beamwidth are so large (given are 3.8+/-0.8 deg and 1.9+/-0.8 deg). If the design of the antenna system is known, it should be possible to calculate the beam width with an accuracy of 15 %. Response: Thank you for your suggestions, I am very sorry for our negligence of the design and dimensions of the antennas and we have added the antenna design description and the schematic antenna structure. The antenna system

contains parabolic reflector (Size:320.5×186.3mm, focal length:180mm), beam splitter and compactness a corrugated feedhorn. Parabolic reflector can focus the beam and be used to scan the beams in elevation. Corrugated feedhorn offers a wide bandwidth, low cross polarization level, low sidelobe level and a rotationally symmetric beam. The MWR have many channels to observe atmospheric radiation intensity in K-band (22-30GHz) and V-band (51-59GHz). The designed beamwidth is less than 5° in K-band and 3° in V-band.

The description above the antenna of MWR has been added to the manuscript. Please see Pages 2 - 4 of the revised version.

4. Comment: Lack of motivation for meteorological applications of MWR. For main applications of radiometers in meteorology, which are temperature and humidity profiling and integrated water vapor, the atmosphere is often assumed to be uniform, i.e. the radiation is constant within the antenna beam. Taking into account that beamwidths of MWR are in the order of 1-4 deg, in the case of clear sky this assumption is fully justified. In such cases only the antenna loss, which is included in the total loss during the hot-cold calibration, matters. Gain and beamwidth have no impact. In presence of liquid clouds, the radiation within the beam is not homogeneous. But it is not clear how more precise knowledge of the beamwidth and gain can help to take this effect into account.

Response: Special thanks for this comment. The manuscript has been revised based on the following description:

As most observations for meteorology are done in the zenith direction, relatively large beamwidths are acceptable. However, this becomes important when viewing at low elevation angles during the period for Tipping curve calibrations. The antenna pattern and pointing error are important influential factors for Tipping curve calibration uncertainties (Han and Westwater, 2000 "Analysis and improvement of tipping calibration for ground-based microwave radiometers"). When Tipping curve calibration is enabled,

the radiometer performs a scan from zenith to 20° elevation, the calibration uncertainties increase by increasing beamwidth. If not corrected, this can introduce a bias to the Tipping curve calibrations (Radiometrics MP3000 Microwave Radiometer Performance Assessment. Technical Report –TR29. Version 1.0). In order to improve the observation and Tipping method calibration accuracy of MWR, the antenna pattern of MWR must be accurately measured on site any time necessary. The purpose of this study is to present a simplified solar method so that the method can be applied operationally in the future. On the other hand, we need to check whether the performance of an antenna in field operation complies with the design specification. Furthermore, in case the antenna is very large or the final assembly occurs at the installation site, the traditional method is extremely difficult. Especially the chamber method cannot be used to measure the antenna pattern of a radiometer in field operation. Therefore, we suggest to use the solar method in our study in order to improve the accuracy of a ground-based MWR observation by automatically checking the pointing accuracy together with alignment correction. And this paper presents the solar method to determine the MWR antenna pattern and to calibrate antenna pointing of MWR networks operating in the field.

The description above about the motivation for applications of MWR has been added to the manuscript. Please see the Lines 39 to 57 on Pages 1 - 2 of the revised version.

Thanks again for your kindly comments and suggestions.

All the best,

LEI Lianfa

On behalf of all the authors
* * *

---

## Author Comment (AC2) · 24 Nov 2020

Thank you very much for your letter and sugesstion.

The comments are all valuable and very helpful for revising and improving our paper. We have studied comments carefully, and have made corrections and modification which we hope meet with approval. We have been trying our best to improve the manuscript. Please see the revised version of the manuscript for detail. The following is the one to one correspondence to the comments.

1. Comment: I am missing some motivation for the study: Of course, it is always desired to characterize an instrument as well as possible. However, more motivation on why is it important to know the exact antenna patterns for ground-based microwave radiometers would be necessary to mention (e.g. temperature profiling, considering antenna patterns in radiative transfer modelling for retrieval algorithms, etc.). Response: Special thanks for this comment. The manuscript has been revised based on the following description:

As most observations for meteorology are done in the zenith direction, relatively large beamwidths are acceptable. However, this becomes important when viewing at low elevation angles during the period for Tipping curve calibrations. The antenna pattern and pointing error are important influential factors for Tipping curve calibration uncertainties (Han and Westwater, 2000 “Analysis and improvement of tipping calibration for ground-based microwave radiometers”). When Tipping curve calibration is enabled, the radiometer performs a scan from zenith to 20° elevation, the calibration uncertainties increase by increasing beamwidth. If not corrected, this can introduce a bias to the Tipping curve calibrations (Radiometrics MP3000 Microwave Radiometer Performance Assessment. Technical Report –TR29. Version 1.0). In order to improve the observation and Tipping method calibration accuracy of MWR, the antenna pattern of MWR must be accurately measured on site any time necessary. The purpose of this study is to present a simplified solar method so that the method can be applied operationally in the future. On the other hand, we need to check whether the performance of an antenna in field operation complies with the design specification. Furthermore, in case the antenna is very large or the final assembly occurs at the installation site, the traditional method is extremely difficult. Especially the chamber method cannot be used to measure the antenna pattern of a radiometer in field operation. Therefore, we suggest to use the solar method in our study in order to improve the accuracy of a ground-based MWR observation by automatically checking the pointing accuracy together with alignment correction. And this paper presents the solar method to determine the MWR antenna pattern and to calibrate antenna pointing of MWR networks operating in the field.

We have added the antenna design description and the schematic antenna structure. The antenna system contains parabolic reflector (Size:320.5×186.3mm, focal length:180mm), beam splitter and compactness a corrugated feedhorn. Parabolic reflector can focus the beam and be used to scan the beams in elevation. Corrugated feedhorn offers a wide bandwidth, low cross polarization level, low sidelobe level and a rotationally symmetric beam. The MWR have many channels to observe atmospheric radiation intensity in K-band (22-30GHz) and V-band (51-59GHz). The designed beamwidth is less than 5° in K-band and 3° in V-band.

The description above the antenna of MWR has been added to the manuscript. Please see Lines 39 - 102 on Pages 1 - 4 of the revised version.

2.Comment: The sun has been used as target for antenna pattern studies before. Please make clear what your novelty is. Would it be possible to apply your method automatically to check the pointing accuracy as well as the alignment of a radiometer that is deployed in the field?

Response: Thanks a lot for guidance. We have noticed that several researchers in literature used the sun to measure weather radar antenna pattern. In this paper, our study is to accurately measure the antenna pattern of MWR with a simplified solar method. We apply this method to measure the MWR pattern and this method can be used to improve the Tipping curve calibration accuracy, automatically check and calibrate the pointing accuracy. Therefore, this method can be used for MWR antenna measurements and hopefully to monitor the antenna pattern and pointing of a radiometer in operational, field applications after installation. It is possible to apply our method automatically to check the pointing accuracy as well as the alignment of a radiometer that is deployed in the field and this is a good idea. We suggest to use the solar method in our study in order to improve the accuracy of a ground-based MWR observation by automatically checking the pointing accuracy together with alignment correction.

The description above about the motivation for applications of MWR has been added

to the manuscript. Please see the lines 39 to 57 on Pages 1 - 2 of the revised version.

3.Comment: The basics on microwave radiometry and antennas cover too much of the manuscript. Most of the information can be found in textbooks. On the other hand, the technical description of the instrument is very poor in terms of receiver technique and other components. How is elevation and azimuth scanning performed by the instrument? What are the sources of uncertainty in this regard? What is the temporal resolution of the brightness temperature measurements when performing scans?

Response: Thank you for your suggestions, I am very sorry for our negligence of the system performance of the instrument. And we have added some technical description of the instrument (receivers, schematic internal and antenna structure of MWR).

During the elevation and azimuth scanning, the antenna is moved to raster a box around the actual sun position using several PPIs, by adding a step value $\Delta\theta$ (-10°~10°) on the elevation and azimuth of the sun in each antenna pointing angle and the observing angle are sent to the antenna servo control system so that the antenna beam can scanning the sun. This scanning can last about 30 minutes at each frequency. Since the sun is moving along the sky within this time interval, the scanning box also follows the sun. The sources of uncertainty contain atmospheric refraction, the fluctuation of solar radiation brightness temperature and the accuracy of the antenna servo control system.

The description above about the motivation for applications of MWR has been added to the manuscript. Please see Lines 65 - 102 on Pages 2 - 4 of the revised version.

4.Comment: What about the repeatability of the method? Did you perform several scans under different conditions and/or solar elevation angles (e.g. summer/winter, morning/noon)? What are your recommendations in that respect?

Response: Thanks for your suggestion, this method can be repeated and we have revised this manuscript. Scans under different solar elevations and seasons have been

completed in order to study the effect of the solar elevation and season variation to the measurement of antenna pattern. Therefore, more observations and results have been added as one can see at Pages 8 - 9 of the revised manuscript. During the elevation and azimuth scanning, the sun at low elevation (<25°) should be avoided because of atmospheric refraction.

Thanks again for your kindly comments and suggestions.

All the best,

LEI Lianfa

On behalf of all the authors

---

## Author Comment (AC3) · 24 Nov 2020

Thank you very much for your letter and suggestion.

The comments are all valuable and very helpful for revising and improving our paper. We have studied comments carefully, and have made corrections and modification which we hope meet with approval. We have been trying our best to improve the manuscript. Please see the revised version of the manuscript for detail. The following is the one to one correspondence to the comments.

1. Comment: The method is not novel, nor is the application novel. A potential novelty

needs to be elaborated in detail and the benefit for MWR measurements have to be highlighted.

Response: Thanks for your suggestion, may be the former version didn't describe the innovative points prominently. We have noticed that several researchers in literature used the sun to measure weather radar antenna pattern, but our study is to accurately measure the antenna pattern of MWR with a simplified method. We apply this method to measure the MWR pattern.

As most observations for meteorology are done in the zenith direction, relatively large beamwidths are acceptable. However, this becomes important when viewing at low elevation angles during the period for Tipping curve calibrations. The antenna pattern and pointing error are important influential factors for Tipping curve calibration uncertainties (Han and Westwater, 2000 "Analysis and improvement of tipping calibration for ground-based microwave radiometers"). When Tipping curve calibration is enabled, the radiometer performs a scan from zenith to $20°$ elevation, the calibration uncertainties increase by increasing beamwidth. If not corrected, this can introduce a bias to the Tipping curve calibrations (Radiometrics MP3000 Microwave Radiometer Performance Assessment. Technical Report –TR29. Version 1.0). In order to improve the observation and Tipping method calibration accuracy of MWR, the antenna pattern of MWR must be accurately measured on site any time necessary. The purpose of this study is to present a simplified solar method so that the method can be applied operationally in the future. On the other hand, we need to check whether the performance of an antenna in field operation complies with the design specification. Furthermore, in case the antenna is very large or the final assembly occurs at the installation site, the traditional method is extremely difficult. Especially the chamber method cannot be used to measure the antenna pattern of a radiometer in field operation. Therefore, we suggest to use the solar method in our study in order to improve the accuracy of a ground-based MWR observation by automatically checking the pointing accuracy together with alignment correction. And this paper presents the solar method to determine the MWR

antenna pattern and to calibrate antenna pointing of MWR networks operating in the field.

The description above about the motivation for applications of MWR has been added to the manuscript. Please see Lines 39 - 57 on Pages 1 - 2 of the revised version.

2. Comment: Technical detail of the MWR model are missing: antenna size, frequencies in K- and V-band, number of receivers

Response: I am very sorry for our negligence of the system performance of the MWR. We have added the following into the revised version.

The antenna system of MWR contains parabolic reflector (Size:320.5×186.3mm, focal length:180mm), beam splitter and compactness a corrugated feedhorn. Parabolic reflector can focus the beam and be used to scan the beams in elevation. Corrugated feedhorn offers a wide bandwidth, low cross polarization level, low sidelobe level and a rotationally symmetric beam. The MWR have two receivers (K- and V-receiver), many channels to observe atmospheric radiation intensity in K-band (22-30GHz) and V-band (51-59GHz), the frequency of observation are 22.235, 22.5, 23.035, 23.835, 25.0, 26.235, 28.0, 30.0, 51.25, 51.76, 52.28, 52.8, 53.34, 53.85, 54.4, 54.94, 55.5, 56.02, 56.66, 57.29, 57.96, 58.8 GHz.

The description above about the motivation for applications of MWR has been added to the manuscript. Please see Pages 2 - 4 of the revised version.

3. Comment: Equations for sun azimuth and elevation are not necessary since they don't describe the necessary information as long as the declination is not given.

Response: Thanks for your suggestion. I have revised this manuscript according to the suggestions.

4. Comment: Too many details in the sections about "model of atmospheric TB" and "model of the antenna power pattern". Why is the opacity of the atmosphere relevant for this study? Minor remarks.

Response: Thanks for your suggestion, I have revised this manuscript. We have described briefly in the sections about "model of atmospheric TB" and "model of the antenna power pattern".

In order to calculate the TB increment of the solar radiation arriving at the antenna without atmospheric attenuation, we must calculate the opacity of the atmosphere in real time. For antenna elevation angles different, the slantwise attenuation is proportional to the ratio of the zenith attenuation for horizontally homogeneous sky conditions.

5. Comment: English grammar and spelling need to be checked. There are many recurrences of statements.

Response: I have carefully corrected them.

6. Comment: The DOI links in the references have wrong syntax: should be https://doi.org/ or doi: instead of https://doi:

Response: The DOI links in the references have been corrected in the manuscript.

7. Comment: Line 38-39: check English grammar

Response: The grammar errors have been checked and corrected.

8. Comment: Line 44: Laura et al., 2017 is missing in references

Response: The missing reference has been checked and corrected.

9. Comment: Lines 48-50: check English grammar

Response: The grammar errors have been checked and corrected.

10. Comment: Lines 51 - 71: several recurrences of statements

Response: I have re-written this part according to the suggestions.

11. Comment: Figure 2 is not necessary, this is not a paper about calibration

Response: I have removed Figure 2.

12. Comment: Figure 3 is not necessary, the solid angle of the sun and estimated solid angle of the antenna give sufficient information to the reader

Response: I have re-written and improved this part according to the suggestions.

13. Comment: Line 140:" only solar emission" What else?

Response: Solar emission and atmospheric emission.

14. Comment: "solid" instead of "sold"

Response: The spelling error has been checked and corrected.

15. Comment: Figure 4: different date as in Figure 5. How can this be related to each other? Not relevant for this manuscript

Response: I have corrected this part according to the suggestions.

16. Comment: Lines 170 - 175: very difficult to understand. How can the antenna be fixed (line 174) during a sun scan (line 174)?

Response: I have re-written this part according to the suggestions. The MWR contains a high-precision elevation and azimuth stepping motor. The antenna pointing angle is sent to the antenna servo control system so that the antenna beam can scanning the sun.

Please see Lines 90 - 102 of the revised version.

17. Comment: Line 173: smaller than 10°

Response: I have corrected this error.

18. Comment: Lines 182-187: I do not understand the context

Response: I have re-written this part. Because of the directional characteristics of the antenna, the TB increment observed is proportional to the ratio of the solar solid angle to the antenna solid angle.

19. Comment: Line 189: Holleman et al.,2010

Response: The missing reference has been corrected.

20. Comment: Line 196: Remove, this has been written very often now

Response: I have removed this part.

21. Comment: Lines 217 - 219: what is the calibration angle? Is it 0 in azimuth and 0 in elevation? Response: It is 0.12° in azimuth and 0.3° in elevation. These data have been added to the revised version of the manuscript.

22. Comment: Figure 7: what are the markers, what are the lines?

Response: I have added the description. The markers are for the observed data and the lines are for fitting of the Gaussian function.

23. Comment: Line 237: what is the H-plane and E-plane in this context? Solar radiation is not polarized

Response: The solar radiation is not polarized but the antenna is. In the case of the linearly polarized antenna, the H-plane is the plane containing the magnetic field vector and the direction of maximum radiation. The E-plane is the plane containing the electric field vector and the direction of maximum radiation. For a vertically polarized antenna, the H-plane usually coincides with the horizontal/azimuth plane. For a horizontally polarized antenna, the H-plane usually coincides with the vertical/elevation plane.

24. Comment: Line 254: what is D?

Response: It is the antenna gain. I have corrected this error.

25. Comment: Line 262 Holleman et al., 2010

Response: The missing reference has been corrected.

26. Comment: calculated

[Figure]

Response: The spelling error has been corrected.

27. Comment: Table 2: it would be necessary to have more than one measurement series to get statistical relevant accuracy of the antenna parameters

Response: I have revised this part according to the suggestions.

28. Comment: Line 297: Why should one use a MWR to observe the radiation of the sun? What will be the context? What will be the difference to the current manuscript?

Response: In this paper, we measured the antenna pattern of the MWR by observing the sun. We hope to observe the variation of solar microwave radiation through long-term observation by the MWR in the future in order to study the effect of the earth's orbital eccentricity on incident solar flux.

Thanks again for your kindly comments and suggestions.

All the best,

LEI Lianfa

On behalf of all the authors